# Kinematic Alignment Achieves a More Balanced Total Knee Arthroplasty Than Mechanical Alignment among CPAK Type I Patients: A Simulation Study

**DOI:** 10.3390/jcm13123596

**Published:** 2024-06-19

**Authors:** Noriaki Arai, Seikai Toyooka, Hironari Masuda, Hirotaka Kawano, Takumi Nakagawa

**Affiliations:** Department of Orthopaedic Surgery, Teikyo University School of Medicine, Tokyo 173-8606, Japan; no3noriaki0729@gmail.com (N.A.);

**Keywords:** functional alignment, kinematic alignment, mechanical alignment, robotic-assisted total knee arthroplasty, soft tissue balance

## Abstract

**Background**: There is no consensus on whether mechanical alignment (MA) or kinematic alignment (KA) should be chosen for total knee arthroplasty (TKA) for coronal plane alignment of the knee (CPAK) Type I with a varus arithmetic HKA (aHKA) and apex distal joint line obliquity (JLO). The aim of this study was to investigate whether MA or KA is preferable for soft tissue balancing in TKA for this phenotype. **Method**: This prospective cohort study included 64 knees with CPAK Type I osteoarthritis that had undergone cruciate-retaining TKA. Using optical tracking software, we simulated implant placement in the Mako system before making the actual bone cut and compared the results between MA and KA. Extension balance (the difference between medial and lateral gaps in extension) and medial balance (the difference in medial gaps in flexion and extension) were examined. These gap differences within 2 mm were defined as good balance. Achievement of overall balance was defined as an attainment of good extension and medial balance. The incidence of balance in each patient was compared with an independent sample ratio test. **Results**: Compared with the MA group, the KA group achieved better soft tissue balance in extension balance (*p* < 0.001). A total of 75% of the patients in the KA group achieved overall balance, which was greater than the 38% achieved in the MA group (*p* < 0.001). **Conclusions**: In robot-assisted TKA for CPAK Type I osteoarthritis, KA achieved knee balance during extension without soft tissue release in a greater percentage of patients than MA.

## 1. Introduction

Two important factors must be considered for successful total knee arthroplasty (TKA): alignment and balance [1,2]. These factors are highly interrelated, and alignment is known to exert significant influence on achieving balance. In recent decades, the mechanical alignment (MA) method has been the gold standard for TKA alignment. The distal femur and proximal tibia are cut perpendicular to their respective functional axes in this method, whereby the mechanical axis passes through the middle of the knee joint and the hip–knee–ankle (HKA) angle is restored to neutral alignment. The implant is placed perpendicular to the osteotomy and is theoretically loaded vertically. The effect of vertical loading has enabled even older generation implants to achieve long-term durability [3]. However, this method ignores the natural alignment of patients to place implants uniformly in all patients. A large international registry survey revealed that 20% of patients were unsatisfied after TKA surgery, and it is speculated that uniform alignment using the MA method may be one of the causes of patient dissatisfaction [4,5,6,7]. Howell et al. introduced the conceptual framework of kinematic alignment, which aims to reproduce the innate “constitutional alignment” of the knee to address the individual differences in knee joint morphology [8,9,10,11]. The use of kinematic alignment is expected to result in more natural knee motion, better soft-tissue balance, and increased patient satisfaction. Currently, there are many reports showing that KA provides better outcomes than MA [12,13,14,15,16].

MacDessi et al. [17] described the coronal plane alignment of the knee (CPAK), a groundbreaking classification system for coronal plane alignment of the knee joint. Two values, the arithmetic HKA (aHKA) and JLO, were used to classify knee alignment and joint inclination into nine phenotypes. The CPAK classification has become an essential part of the KA method, as it can be used to estimate constitutional alignment even in patients with knee osteoarthritis. Varus aHKA with a medially tilted articular surface is classified as CPAK Type I, and the phenotype has been reported to be the most common in Asia [18,19,20]. Although reports have generally suggested that KA leads to a more balanced knee than MA, there are no reports indicating whether KA is superior to MA in Type I patients, which is a phenotypical characteristic that is particular to specific regional populations in terms of balance. Therefore, the purpose of this study was to determine whether KA or MA can achieve balance in CPAK Type I patients, which is the most common in Asia. The hypothesis of this study was that KA can achieve better soft tissue balance than can MA among CPAK Type I patients.

## 2. Materials and Methods

### Patients and Design

The study protocol was approved by the institutional review board of [Teikyo University], and all patients provided informed consent.

Data for 127 knees of 88 consecutive patients with knee osteoarthritis who underwent primary TKA between January 2022 and June 2023 at a single institution were analyzed prospectively. All patients were of Asian descent. Among these patients, only those with CPAK Type I were included in this study. Type I patients with an alignment of at least 2° varus aHKA and apex distal JLO of less than 177° medially were selected according to a method described by MacDessi et al. [17] using medial proximal tibial angle (MPTA) and lateral distal femoral angle (LDFA) measurements from preoperative standing whole-leg radiographs. The exclusion criteria for twelve patients were osteonecrosis, posterior cruciate ligament (PCL) dysfunction, severe range of motion limitation (flexion contracture greater than 30°), rheumatoid arthritis, presence of extra-articular deformity of the femur or tibia, prior osteotomy of the femur or tibia, prior intra- or extra-articular fracture of the femur or tibia, and prior hip replacement surgery.

Patients underwent a preoperative CT scan to identify their hip and ankle centers. CT scans were subsequently used to develop three-dimensional models of the knees of the patients. During the TKA procedure, the balance of implant placement was simulated using the Mako planning software (Version 2.0, Stryker, Fort Lauderdale, FL, USA) prior to the actual osteotomy. The Mako software provided bone resection plans for the preferred implant positioning. The alignment, resection depth, and size of the implants were modified and optimized according to the preferences of the surgeon. To achieve the appropriate balance, the MA and KA plans were applied to the same knee of the patient, the balance after implant placement was evaluated, and the difference in balance obtained by fine-tuning the implant position. Each plan was subsequently compared in a simulation.

All TKAs were performed by three board-certified orthopedic surgeons with over 11 years of experience under the supervision of one expert orthopedic surgeon. During the surgical procedure, trans-osseous pins were first inserted into the tibia and femur. Navigation tracking arrays were subsequently attached. All patients underwent surgery via a medial parapatellar approach while preserving as much of the medial soft tissue as possible. The anterior cruciate ligament (ACL) was resected prior to balancing. Bony landmarks of the knee were registered and validated in the Mako planning software using the three-dimensional CT model. After removal of the osteophytes, the soft tissue balance was evaluated with the planned position of the component by exerting varus and valgus stresses to obtain maximal gaps. Measurements were performed at 10 degrees extension and 90 degrees flexion while varus and valgus stress were applied to the knee. The planning software was used to virtually measure the gaps between bone cuts to the nearest mm.

The planning parameters for femoral bone resection of MA knees were set as follows: 6.5 mm from the most distal and posterior points of the femoral condyles to account for the estimated 2 mm thickness of the cartilage and thickness of the implant, neutral alignment, parallel to the surgical trans-epicondylar axis (SEA), and 0 to 7° flexion based on optimal sizing of the femoral component. Tibial bone resections were set to 6.5 mm off the higher side to account for the thickness of the implant, neutral coronal alignment, rotation set to Akagi’s line, and 0 to 7° of posterior slope based on the native lateral tibial plateau. Following an assessment of balance in the initial plan, the component positioning was adjusted to functional alignment, which aims to obtain equal extension and flexion of the medial gaps of the knee. Specifically, to minimize medial and lateral gaps in extension and medial gaps in extension and flexion, the position and size of the implants were adjusted with reference to the gap value after implantation, which was assessed by manually applying varus and valgus stress. Specifically, only the amount of osteotomy, implant size, and femoral flexion angle were adjusted without changing the osteotomy alignment perpendicular to the functional axis or rotational alignment.

The planning parameters of the KA knees were restricted to a boundary of 4° varus to 2° valgus on the tibia and 4° valgus to 2° varus on the femur. Osteotomy angles in the coronal plane of the tibia and femur were determined from the preoperative CPAK classification, and those exceeding the boundaries were controlled to lie within the predetermined limits. The osteotomy plane was planned with a 6.5 mm osteotomy from the most distal region of the medial or lateral femoral condyle to account for the thickness of the implant. Femoral rotation was set parallel to the posterior condylar axis but within 4° of internal or external rotation from the SEA. Femoral flexion was adjusted between 0 and 6° based on the optimal femoral component size. Tibial resection was used to determine the osteotomy plane at 6.5 mm from the most proximal point to account for the thickness of the implant. The posterior slope was set between 0 and 7° based on the native lateral tibial plateau, and rotation was set to Akagi’s line. Following an assessment of balance with initial KA planning, component positioning was adjusted to functional alignment, an alignment technique to optimize soft tissue balance. As with KA, the implant position and size were adjusted based on the gap values after implant placement evaluated by manually applying varus and valgus stresses. The distal femoral and proximal tibial osteotomy alignment, osteotomy volume, implant size, and femoral flexion angle were also adjusted within boundaries to minimize medial and lateral gaps in extension and medial gaps in extension and flexion. The component alignment boundaries in KA are shown in Table 1.

Balance in the present study was determined according to the assessment described by Clark et al. [21]. The aim of achieving a balanced TKA through conventional means such as measured resection or gap balancing techniques has been to ensure that the medial and lateral extension and flexion gaps are equal. According to previous studies, however, the native knee is not symmetrically balanced [22,23]. Several reports have discounted the need for a balanced lateral flexion gap and described the importance of reproducing the natural variation in the lateral flexion gap by providing more laxity in lateral flexion to achieve improved kinematics [24,25,26].

This study assessed the extension balance (difference between medial and lateral gaps at extension), medial balance (difference between medial extension and medial flexion gaps), and overall balance of the knee. Extension balance was defined in the simulation software (Version 2.0, Stryker Mako, Fort Lauderdale, FL, USA) as the difference between the lateral component gap under varus stress and the medial component gap under valgus stress in the extended position. Measurements were taken in 10-degree flexion, because the fully extended position is affected by the posterior capsule. In addition, medial balance was defined as the difference between the medial extension gap and the medial flexion gap under manual valgus stress. Although equal gaps were the preferential target, the extension balance and medial balance were considered balanced when the assessed gaps were within 2 mm of each other. The overall balance was considered balanced when both extension balance and medial balance were obtained. The extension, medial, and overall balance of the knees were compared between the KA and MA groups.

The *t*-test and chi-square test were used to examine differences between the two groups. The statistical significance level was set at *p* = 0.05, and all calculations were performed using SPSS version 12 (SPSS Inc., Chicago, IL, USA). Knee balance counts in the MA and KA groups were compared between the two groups using the chi-square test. Coronal alignment surgical plans with MA and KA were compared by *t*-test.

## 3. Results

### 3.1. Patient Demographics

One hundred twenty-seven knees of 88 consecutive patients were included in this study. Of these, 51 knees of 43 patients did not require CPAK Type I and were thus excluded from the study. Four patients with bone loss, 3 with severe flexion contracture, and 5 with PCL dysfunction were excluded. The remaining 64 TKAs were included in this study. The percentage of CPAK Type I was 59.8%, similar to that reported by Toyooka et al. [20] The flow chart of patient selection is illustrated in Figure 1. Patient demographics are shown in Table 2.

The mean patient age was 78.5 ± 7.3 years, and the majority of the patients were female. All included knees were CPAK Type I, aHKA was varus, and the joint line obliquity (JLO) was apex distal. The mean aHKA was −5.1 ± 2.6°, and the average JLO was 172.0 ± 2.6°.

### 3.2. Extension Balance, Medial Balance, and Overall Balance

Extension balance, medial balance, and overall balance with KA and MA are shown in Table 3. In terms of medial balance, 100% of the MA knees and 94% of the KA knees achieved balance, and both groups had a high rate of achievement (*p* = 0.01). An extension balance was achieved in 83% of the KA knees and only 38% of the MA knees (*p* < 0.001). Overall balance was also observed in 75% of the KA knees compared to only 38% of the MA knees (*p* < 0.001). A scatter plot demonstrating the balance of knees with different alignment plans is shown in Figure 2. This balance was maintained after prosthesis implantation with no significant difference in balance.

### 3.3. The Surgical Plan Alignment

The surgical plan (Table 4) showed significant differences in all alignment parameters between MA and KA. In MA, the components were created almost neutrally since the alignment was not changed. In KA, the valgus angle of the femur, the varus angle of the tibia, and femoral rotation were smaller than those innate to the patient due to restricted angulation.

## 4. Discussion

The most important finding of this study is that, in CPAK Type I knees, KA requires less knee soft tissue alteration than MA and can provide a more balanced TKA. If an imbalance of more than 2 mm is observed in this phenotype, which is common in Asian populations, the knee requires soft tissue release to achieve proper balancing of the TKA.

A medial balance was obtained in all patients with MA and 94% with KA. In the present study, simulation software was used to adjust the osteotomy volume to achieve as good a gap as possible. By using this procedure, the difference between the medial extension and flexion gaps might be minimized, and medial balance could have been achieved at high rates in both groups. In actual clinical practice without simulation software, a medial balance can be obtained for both MA and KA by adjusting the remaining osteotomy after creating an extension or flexion gap. The MA technique uses the most prominent joint surface (primarily the medial femoral condyle and lateral tibial plateau) as a reference for resection thickness when making the extension gap, and the technique tends to reduce the extension of the medial gap in varus knees and the lateral gap in valgus knees. With the extension gap, however, a significant balance was obtained in KA at 83% compared with MA at 38%. If the MA alignment is maintained, it would be difficult to achieve balance no matter how the implant is adjusted. In varus knees, a perpendicular cut to the mechanical axis results in a gap that is inevitably more open on the lateral side in the coronal plane since the original varus alignment is not maintained [27].

There have been some reports that a more adequate balance can be obtained with KA than with MA. Blakeney et al. [28] compared MA and KA osteotomy simulations using CT to examine the amount of osteotomy and the resulting extension–flexion gap and lateral–medial gap. They reported that KA resulted in a greater percentage of balanced knees than MA. MacDessi et al. [29] also conducted a randomized controlled trial in which they inserted pressure sensors inside the TKA and compared the differences in medial and lateral pressure between the MA and KA. The mean inter-fossa pressure differences evaluated at 10° of flexion, 45° of flexion, and 90° of flexion were all greater in the KA group. However, these data include patients with all knee joint alignments, and there are no studies in the literature that mention CPAK Type I only. CPAK Type I, in which the aHKA is varus with an apex distal to the JLO, is the most common classification in the Asian population. The knees studied in this study had a mean LDFA of 88.5° and an MPTA of 83.5°. The varus constitutional characteristic of the tibia had a greater impact on the overall varus alignment than did the femoral morphology. When the tibia is cut perpendicular to the mechanical axis in the MA, the lateral side of the tibia is cut to a greater extent than the medial side, resulting in loose lateral soft tissue tension. On the other hand, when the proximal tibia is cut varus with KA, the lateral compartment gap becomes less loose. The difference in the proximal tibial bone cutting line is assumed to have greatly influenced the present results, in which more extension balance was obtained in the KA than in the MA.

Clark et al. [21] also reported soft tissue balance using the same software and in a similar manner to the present study. There are some differences between their reported data and ours. First, the extension balance in the MA group was 38% in our study, whereas Clark et al. reported a higher percentage of the study population (84%). Furthermore, the overall balance in MA was 38% in our study, whereas that in their study was 55%. Differences in the alignment of patients may be a contributing factor to this difference in results. In general, the most common phenotype in the Caucasian population is CPAK Type 2, in which the aHKA is neutral with an apex distal JLO. In the present study, the data comprised CPAK Type I patients with a strong varus tilt of the tibia, which may have made it more difficult to achieve extension balance by cutting with MA. Second, the overall balance, extension balance, and medial balance in the KA group were 75%, 83%, and 94%, respectively, in our study, whereas their study was even greater at 97%, 99% and 97%, respectively. Although there was no difference in medial balance in the KA between the two studies, the overall balance differed due to the significant difference in extension balance. An extension balance was not achieved in these patients with extremely reduced MPTA. In addition to the aforementioned differences in target alignment, another possible factor is the difference in alignment boundaries. Our component alignment boundary in KA was 4°, whereas their limit was 6°. It is possible that the difference between studies could have been reduced if the alignment boundaries had widened. Greater boundaries of component alignment may accommodate patients with greater tibial varus and provide a better extension gap.

This study has several limitations. The main weakness of this study is that it is the result of a pre-osteotomy software simulation rather than a postoperative balance assessment with actual implant insertion. However, the software is reliable and has been validated in a number of papers [21,30,31]. In this study, the post-implant placement balance achieved with robot-assisted TKA (RATKA) planning software was virtually equivalent to the balance achieved with pre-osteotomy balancing. Chang et al. also demonstrated that the assessment of the soft tissue gap before osteotomy using this software was comparable and consistent with the actual post-implant gap [32]. The fact that the simulation allowed comparisons to be made for the same knee in the same patient is a strength of this study. Furthermore, virtual measurements from the same software package were used in place of postoperative measurements to compare MA and KA. As the gap measurements obtained were manual, it was not possible to quantify the forces used. This effect was minimized using consistent techniques by the same surgeon on the same patient. Second, the study was not a randomized trial but rather a sequential cohort study.

## 5. Conclusions

In RA TKA for CPAK Type I osteoarthritis, KA achieved a balanced knee, especially in extension, without soft tissue release in a greater percentage of patients than MA.

## Figures and Tables

**Figure 1 jcm-13-03596-f001:**
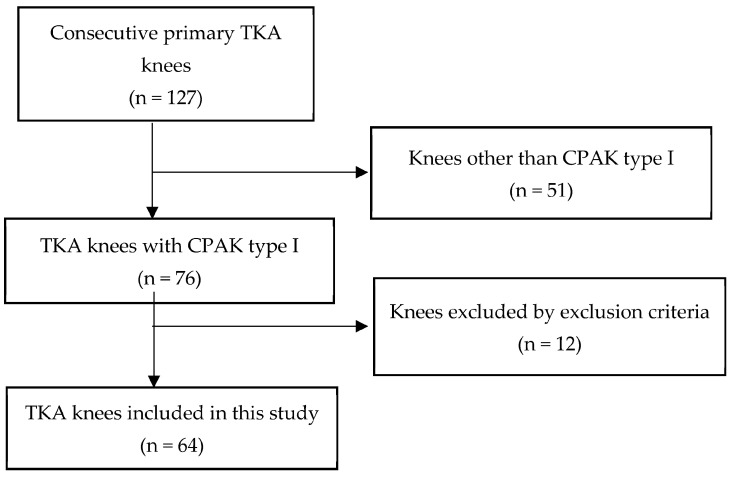
Patient flow chart.

**Figure 2 jcm-13-03596-f002:**
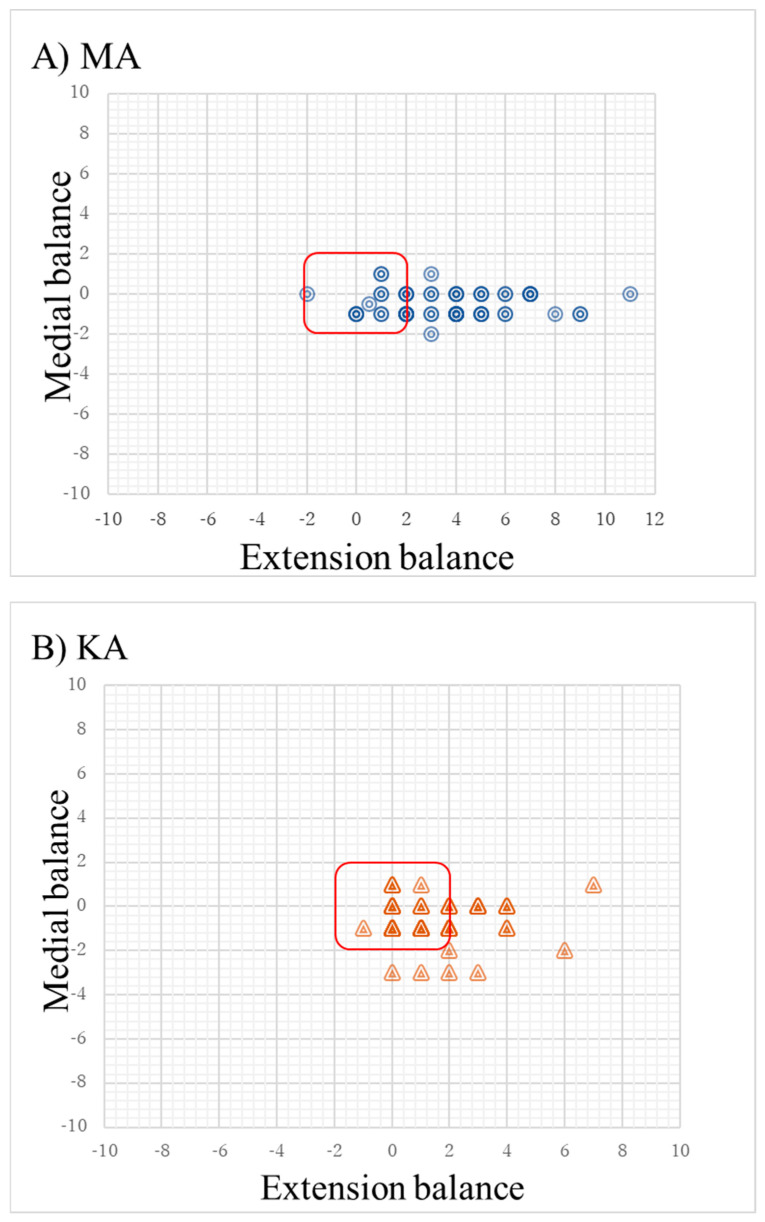
Scatter plot of knee balance according to different alignment plans. (**A**) Mechanical alignment plan (MA). (**B**) Kinematic alignment plan (KA). The *x*-axis shows extension balance (difference between medial and lateral extension gaps at 10° flexion [mm]), and the *y*-axis shows medial balance (difference between medial extension and medial flexion gaps [mm]). The red center box indicates the predetermined limit within 2 mm of balance. The color is darkened in areas where there are multiple patient balances on the same site. The more there are, the darker the color.

**Table 1 jcm-13-03596-t001:** Final component alignment boundaries in KA (varus negative, valgus positive).

Coronal Limits	
Hip–knee–ankle angle	−4° to + 4°
Femoral component	+2° to −4°
Tibial component	−4° to +2°
Sagittal limits	
Femoral flexion	0–7°
Tibial slope	0–7°

**Table 2 jcm-13-03596-t002:** Patient demographics.

Characteristics	
n	64 knees
Number of patients	47
Gender ratio (male:female)	15:32
Right-to-left ratio	28:36
Age	78.5 ± 7.3 years
Height	152.5 ± 7.8 cm
Weight	58.7 ± 10 kg
BMI	25.0 ± 4.0 kg/m^2^
aHKA	−5.1 ± 2.6° (−1~−12°)
JLO	172.0 ± 2.6° (166~177°)
Mean MTPA	83.5 ± 1.9° (78~87°)
Mean LDFA	88.5 ± 2.0° (84~92°)

**Table 3 jcm-13-03596-t003:** Number of knees balanced with MA and KA with percentage of cohort in brackets. The two groups were compared using the chi-square test.

	Balance Achieved	MA 64 Knees	KA 64 Knees	*p* Value
Extension balance	≦±1 mm	11 (17%)	37 (58%)	<0.001
	≦±2 mm	24 (38%)	53 (83%)	<0.001
Medial balance	≦±1 mm	63 (98%)	58 (91%)	=0.1
	≦±2 mm	64 (100%)	60 (94%)	=0.1
Overall balance	≦±1 mm	11 (16%)	35 (55%)	<0.001
	≦±2 mm	24 (38%)	48 (75%)	<0.001

**Table 4 jcm-13-03596-t004:** Coronal alignment surgical plan with MA and KA. The two groups were compared using a *t*-test.

	MA	KA	*p* Value
Femoral coronal alignment	0.0°	0.8 ± 0.9° (−3.5~0)	<0.001
Tibial coronal alignment	0.0°	−3.8 ± 0.6° (−4~0.5)	<0.001
aHKA	0.0°	−3.0 ± 1.1° (−4~0)	<0.001
Joint line obliquity	180.0°	177.5 ± 1.1° (173~179)	<0.001
Femoral rotation to SEA	0°	−0.8 ± 1.8° (−4~4.5)	<0.001

## Data Availability

The original contributions presented in the study are included in the article, further inquiries can be directed to the corresponding author.

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
