# Peer review of "Kinematic Alignment Achieves a More Balanced Total Knee Arthroplasty Than Mechanical Alignment among CPAK Type I Patients: A Simulation Study"

_jcm, 2024, doi:10.3390/jcm13123596_

Round 1

Reviewer 1 Report

Comments and Suggestions for Authors

Dear Sir or Madam,

 1. GENERAL COMMENTS

Thank you for submitting the manuscript entitled “Kinematic alignment achieves a more balanced total knee arthroplasty than mechanical alignment among CPAK Type I patients: a simulation study" to Journal of Clinical Medicine. This study involves a simulation study in kinematic alignment total knee arthroplasty and mechanical alignment among CPAK Type I patients. The findings of the study indicate that KA achieved knee balance during extension without soft tissue release in a greater percentage of patients than MA. The overall structure of the article is complete, focusing on patients with CPAK type I knee osteoarthritis based on existing research. However, there are still some questions regarding the article. Therefore, I want to offer some suggestions.

2. TITLE

The title is fine.

3. ABSTRACT

The abstract is well-structured and clear.

Line 12: Please clarify whether this paper is a prospective study or a retrospective study, as there is a discrepancy with the statement in line 67.

4. INTRODUCTION

The Introduction section is well written. However, including all the content in a single paragraph seems lengthy. Have the authors considered separating it into different sections?

Line 52-54: The authors mention that CPAK I is the most common phenotype. The reviewer believes this might be inaccurate. Please verify this statement, as CPAK type II appears to be more prevalent than type I, except for one study.

 Line 55-56: Although the author mention "no reports indicating whether KA is superior to MA in Type I patients," there is a lack of explanation for selecting the Type I population. Please discuss why the Type I population was chosen to provide sufficient background for the readers.

5. MATERIALS AND METHODS

The description in the Materials and Methods section is comprehensive.

Line 66-67 The primary concern lies in whether this article is a prospective study or a retrospective one.

 Line 127-133: This section does not seem to belong here.

 Line 135-145: Please ask the authors to provide a detailed description of the software used and how the assessment of "balance" was conducted.

6. RESULTS

In general, the Results section comprehensively elucidates the primary findings of the study.

 Line 153-154 The primary concern regarding the results is that the proportion of CPAK Type I classification by the authors appears to be too high. In general literature, the proportion of CPAK Type I classification does not seem to exceed 30%.

 Line 170-171: The results lack the actual postoperative imaging findings. The reviewer would like to know if the surgery was performed according to the preoperative plan and what the outcomes were.

 Line 174-176: Please explain the meaning of this paragraph. If it is incorrect, kindly delete it.

7. DISCUSSION

Line 240-241: The authors repeatedly change the notation of CPAK type 1/I in the text; please maintain consistency.

 Line 266-267: Data not presented in the results section should not be included in the discussion.

 Line 282-283: The phrase "the simulation allowed comparisons to be made for the same knee in the same patient is a strength of this study" is mentioned in the discussion. However, the reviewer seems not to have found evidence in the article supporting the claim that "comparisons to be made for the same knee in the same patient" were conducted.

8. CONCLUSION

In my opinion, the conclusion is in accordance with the reported results.

Line 290: "Robot-assisted TKA" has already been abbreviated earlier; it should be consistently abbreviated throughout.

9. REFERENCES

The cited references are quite dated; for example, there have been several updates to the meta-analyses since 2018. Please update the references accordingly.

10. FIGURES

Figure 1: The authors should maintain consistent formatting in the figures. In some boxes, (n =?) is recorded, and the reviewer believes that this should be included in the other boxes as well.

Figure 2: A and B are not labeled in the figure; please verify. Additionally, do the varying shades of the circles in the figure have any meaning?

11. TABLES

Since no “Abbreviation List” has been provided at the beginning, have you considered adding one in the table section to explain the abbreviated names?

Author Response

 Comment 1:

Line 12: Please clarify whether this paper is a prospective study or a retrospective study, as there is a discrepancy with the statement in line 67.

Authors’ Response:

We would like to thank Reviewer #1 for this comment. We added a sentence to clarify that this was a prospective study.

Line 12: This prospective cohort study included 64 knees with CPAK Type I osteoarthritis that had undergone cruciate-retaining TKA.

Comment 2:

The Introduction section is well written. However, including all the content in a single paragraph seems lengthy. Have the authors considered separating it into different sections?

Authors’ Response: We would like to thank Reviewer #1 for this comment. Per suggestion, we have included a section break to split the Introduction section in paragraphs (between Line 48-51).

Line 48-51:

[…] Currently, there are many reports showing that KA provides better outcomes than MA. [12-16]

 MacDessi et al. [17] described the Coronal Plane Alignment of the Knee (CPAK), a groundbreaking classification system for coronal plane alignment of the knee joint.

Comment 3:

Line 52-54: The authors mention that CPAK I is the most common phenotype. The reviewer believes this might be inaccurate. Please verify this statement, as CPAK type II appears to be more prevalent than type I, except for one study.

Authors’ Response:

We agree that this sentence may benefit from further clarification. The prevalence of the phenotype was specific to the geographical location; therefore, we rewrote the sentence to say that Type I was the most common in Asia.

Line 55-56: Varus aHKA with a medially tilted articular surface is classified as CPAK Type I, and the phenotype has been reported to be the most common in Asia.

Comment 4:

Line 55-56: Although the author mention "no reports indicating whether KA is superior to MA in Type I patients," there is a lack of explanation for selecting the Type I population. Please discuss why the Type I population was chosen to provide sufficient background for the readers.

Authors’ Response: We agree that the explanation was insufficient. The high prevalence of the phenotype in the Asian population was the reasoning for selecting the Type 1 population; therefore, we added the following phrase in the manuscript:

Line 60-61: Therefore, the purpose of this study was to determine whether KA or MA can achieve balance in CPAK Type I patients, which is the most common in Asia.

Comment 5:

Line 66-67 The primary concern lies in whether this article is a prospective study or a retrospective one.

Authors’ Response: We would like to thank Reviewer #1 for this comment. We have added the term “prospective” for clarification.

Line 12: This prospective cohort study included 64 knees with CPAK Type I osteoarthritis that had undergone cruciate-retaining TKA.

Line 68-70: Data for 127 knees of 88 consecutive patients with knee osteoarthritis who underwent primary TKA between January 2022 and June 2023 at a single institution were analyzed prospectively.

Comment 6:

Line 127-133: This section does not seem to belong here.

Authors’ Response: We would like to thank Reviewer #1 for this comment. We considered moving line 136 through 143 to the Introduction and Discussion sections; however, since this is the basic premise of this study, we think it is best explained in the Methods section. As noted by the reviewer, moving this section to the Introduction would make the Introduction longer. We cannot find an appropriate place for it in Discussion either. The prior paper by Clark et al. provides a similar explanation in Methods, so we would like to leave this section as-is.

Comment 7:

Line 135-145: Please ask the authors to provide a detailed description of the software used and how the assessment of "balance" was conducted.

Authors’ Response: We would like to thank Reviewer #1 for this comment. We have made changes to the manuscript for clarity.

Line 144-156:

This study assessed the extension balance (difference between medial and lateral gaps at extension), medial balance (difference between medial extension and medial flexion gaps), and overall balance of the knee. Extension balance was defined in the simulation software (Stryker Mako, Fort Lauderdale, FL, USA) as the difference between the lateral component gap under varus stress and the medial component gap under valgus stress in the extended position. Measurements were taken in 10-degree flexion, because the fully extended position is affected by the posterior capsule. In addition, medial balance was defined as the difference between the medial extension gap and the medial flexion gap under manual valgus stress. Although equal gaps were the preferential target, the extension balance and medial balance were considered balanced when the assessed gaps were within 2 mm of each other. The overall balance was considered balanced when both extension balance and medial balance were obtained. The extension, medial, and overall balance of the knees were compared between the KA and MA groups.

Comment 8:

Line 153-154 The primary concern regarding the results is that the proportion of CPAK Type I classification by the authors appears to be too high. In general literature, the proportion of CPAK Type I classification does not seem to exceed 30%.

Authors’ Response: We would like to thank Reviewer #1 for this comment. CPAK Type I is common in Asia. We have added a reference to substantiate this claim.

Line 168-169: The percentage of CPAK Type I was 59.8%, similar to that reported by Toyooka et al [20].

Comment 9:

Line 170-171: The results lack the actual postoperative imaging findings. The reviewer would like to know if the surgery was performed according to the preoperative plan and what the outcomes were.

Authors’ Response: We would like to thank Reviewer #1 for this comment. MA and KA planning were applied for the same knee, and all surgery was performed with KA. As a result, the balance after implant placement was similar to the simulation results. This was noted in the following sentence:

Line184-185: This balance was maintained after prosthesis implantation with no significant difference in balance.

Comment 10:

Line 174-176: Please explain the meaning of this paragraph. If it is incorrect, kindly delete it.

Authors’ Response: We would like to thank Reviewer #1 for bringing this to our attention. We have deleted the instructional sentences from the MDPI template that were accidentally left undeleted.

Line 174-176 (line numbers from original submission): It should provide a concise and precise description of the experimental results, their interpretation, as well as the experimental conclusions that can be drawn.

Comment 11: Line 240-241: The authors repeatedly change the notation of CPAK type 1/I in the text; please maintain consistency.

Authors’ Response: We would like to thank Reviewer #1 for this comment. Per suggestion, we have changed the notation of CPAK for consistency.

Line 270-271: However, these data include patients with all knee joint alignments, and there are no studies in the literature that mention CPAK Type Ⅰ only.

Comment 12: Line 266-267: Data not presented in the results section should not be included in the discussion.

Authors’ Response: We agree that data not presented in the results should not be included in the discussion. We have deleted this sentence.

Line 266-267 (line numbers from original submission): According to our data, 17% of patients were unable to achieve extension balance in the KA, with a mean LDFA of 88.8 ± 1.7° (85-91°) and an MPTA of 82.4 ± 1.2° (78-86°).

Comment 13: Line 282-283: The phrase "the simulation allowed comparisons to be made for the same knee in the same patient is a strength of this study" is mentioned in the discussion. However, the reviewer seems not to have found evidence in the article supporting the claim that "comparisons to be made for the same knee in the same patient" were conducted.

Authors’ Response: We would like to apologize for the lack of clarity. This claim is now described in Line 85-88 of the Methods section.

Line 85-88: To achieve the appropriate balance, the MA and KA plans were applied to the same knee of the patient, the balance after implant placement was evaluated, and the difference in balance obtained by fine-tuning the implant position. Each plan was subsequently compared in a simulation.

Comment 14: Line 290: "Robot-assisted TKA" has already been abbreviated earlier; it should be consistently abbreviated throughout.

Authors’ Response: We would like to thank Reviewer #1 for this comment. We have made the suggested change to the manuscript.

Line 318: In RA TKA for CPAK Type I osteoarthritis, KA achieved a balanced knee, especially in extension, without soft tissue release in a greater percentage of patients than MA.

Comment 15:

The cited references are quite dated; for example, there have been several updates to the meta-analyses since 2018. Please update the references accordingly.

Authors’ Response:

We agree with Reviewer #1’s suggestion. Per suggestion, we have updated the references with the following articles:

1, Oussedik S, Abdel MP, Victor J, Pagnano MW, Haddad FS. Alignment in total knee arthroplasty. Bone Joint J. 2020;102-B(3):276-279

13, An VVG, Twiggs J, Leie M, Fritsch BA. Kinematic alignment is bone and soft tissue preserving compared to mechanical alignment in total knee arthroplasty. Knee. 2019 Mar;26(2):466-476

23, Deep K, Picard F, Clarke JV. Dynamic Knee Alignment and Collateral Knee Laxity and Its Variations in Normal Humans. Front Surg. 2015 Nov 25;2:62

Comment 16:

Figure 1: The authors should maintain consistent formatting in the figures. In some boxes, (n =?) is recorded, and the reviewer believes that this should be included in the other boxes as well.

Authors’ Response: We would like to thank Reviewer #1 for this comment. Per suggestion, we have made changes to the formatting for consistency.

Comment 17:

Figure 2: A and B are not labeled in the figure; please verify. Additionally, do the varying shades of the circles in the figure have any meaning?

Authors’ Response:

We would like to thank Reviewer #1 for this comment. We added the label.

The color is darkened in areas where there are multiple patient balances on the same site. The more there are, the darker the color. We have added this description to the figure legend.

Comment 18: Since no “Abbreviation List” has been provided at the beginning, have you considered adding one in the table section to explain the abbreviated names?

Authors’ Response: We would like to thank Reviewer #1 for this comment. Per suggestion, we have created an abbreviation list at the beginning.

Reviewer 2 Report

Comments and Suggestions for Authors

In this manuscript, Arai et al simulated head-to-head comparison between MA and KA in patients with CPAK Type 1 alignment that underwent MAKO TKA. This study nicely demonstrated that in CPAK Type I knee, KA can achieve predicted balanced gaps in flexion and extension with minimum soft tissue release as compared to MA. This is expected for KA and one of the benefit of doing KA. 

Below are our suggestions for the manuscript:

1. While the author nicely demonstrated in the surgical benefit of KA vs MA, the manuscript would benefit from demonstrating the final flexion-extension balance and medial-lateral balance after the actual final component implantation between MA and KA. This will greatly complement the manuscript and will inform us the correlation between predicted and actual gaps.

2. MacDessi et al, 2021, BJJ, 103-B:329-337 showed that for CPAK type I knee, while KA achieved 100 % balanced knee, MA only achieved balanced in 15 % of patients. It would be interesting to see what the final balance percentage after actual implantation is for these patients' cohort in this manuscript. 

3. In the methods, for KA, while the author explained the boundary of the KA, the author should elaborate on how the surgeon decide on the final varus/valgus angle in the femur and tibia. 

4. Are 127 TKA performed by single surgeon ? The varus/valgus registration can be variable based on the surgeon who performed the registration. 

5. Another supposed benefit of KA is less overall bone cut to achieve balanced flexion and extension. Can the author add the amount of bone cut needed for MA vs KA for the femur and tibia ?

Comments on the Quality of English Language

- The wording in abstract "we simulated the balance after implant placement before making the actual osteotomy" is confusing. I presume the authors meant simulated implant placement in the MAKO system before making the actual bone cut.

- In the result section, the authors left these sentences from the presumed template: This section may be divided by subheadings. It should provide a concise and precise description of the experimental results, their interpretation, as well as the experimental conclusions that can be drawn. Please remove these sentences.

Author Response

Comment 1: While the author nicely demonstrated in the surgical benefit of KA vs MA, the manuscript would benefit from demonstrating the final flexion-extension balance and medial-lateral balance after the actual final component implantation between MA and KA. This will greatly complement the manuscript and will inform us the correlation between predicted and actual gaps.

 Authors’ Response: We would like to thank Reviewer #2 for this comment. The surgery was performed as planned. We have added an explanation for clarification:

 Line 184-185: This balance was maintained after prosthesis implantation with no significant difference in balance.

 Comment 2: MacDessi et al, 2021, BJJ, 103-B:329-337 showed that for CPAK type I knee, while KA achieved 100 % balanced knee, MA only achieved balanced in 15 % of patients. It would be interesting to see what the final balance percentage after actual implantation is for these patients' cohort in this manuscript.

Authors’ Response: We would like to thank Reviewer #2 for this comment. MA and KA planning were applied for the same knee, and all surgery was done with KA. As a result, the balance after implant placement was similar to the simulation results. This was noted.

 Line 184-185: This balance was maintained after prosthesis implantation with no significant difference in balance.

Comment 3: In the methods, for KA, while the author explained the boundary of the KA, the author should elaborate on how the surgeon decide on the final varus/valgus angle in the femur and tibia.

Authors’ Response: We would like to thank Reviewer #2 for this comment. We have added a description of how the final varus/varus angles in the femur and tibia were determined by keeping the medial and lateral gaps in extension and medial gaps in extension and flexion at minimum.

 Line131-134: The distal femoral and proximal tibial osteotomy alignment, osteotomy volume, implant size, and femoral flexion angle were also adjusted within boundaries to minimize medial and lateral gaps in extension and medial gaps in extension and flexion.

Comment 4:

Are 127 TKA performed by single surgeon? The varus/valgus registration can be variable based on the surgeon who performed the registration.

Authors’ Response: We would like to thank Reviewer #2 for this comment. Our apologies. Three orthopedic surgeons performed the surgery under expert management. The description has been added to convey the message.

 Line 89-90: All TKAs were performed by three board-certified orthopedic surgeons with over 11 years of experience under the supervision of one expert orthopedic surgeon.

 Comment 5:

Another supposed benefit of KA is less overall bone cut to achieve balanced flexion and extension. Can the author add the amount of bone cut needed for MA vs KA for the femur and tibia?

Authors’ Response: We would like to thank Reviewer #2 for this comment. Unfortunately, the current study did not investigate the amount of bone cut needed for MA vs KA for the femur and tibia, and we feel this information is outside the scope of our study.

 Comment 6: The wording in abstract "we simulated the balance after implant placement before making the actual osteotomy" is confusing. I presume the authors meant simulated implant placement in the MAKO system before making the actual bone cut.

Authors’ Response: We would like to thank Reviewer #2 for this comment. Per suggestion, we have reworded the sentence for clarity.

 Line14: we simulated implant placement in the MAKO system before making the actual bone cut

 Comment 7:

In the result section, the authors left these sentences from the presumed template: This section may be divided by subheadings. It should provide a concise and precise description of the experimental results, their interpretation, as well as the experimental conclusions that can be drawn. Please remove these sentences.

Authors’ Response: We would like to thank Reviewer #2 for this comment. We have deleted the instructional sentences that were accidentally left undeleted, and we have included subheadings to divide the Results into subsections.

 Line 174-176 (line numbers from original submission):This section may be divided by subheadings. It should provide a concise and precise description of the experimental results, their interpretation, as well as the experimental conclusions that can be drawn.